# Association between Adult Vaccine Hesitancy and Parental Acceptance of Childhood COVID-19 Vaccines: A Web-Based Survey in a Northwestern Region in China

**DOI:** 10.3390/vaccines9101088

**Published:** 2021-09-27

**Authors:** Kezhong A, Xinyue Lu, Jing Wang, Linjie Hu, Bingzhe Li, Yihan Lu

**Affiliations:** 1Institute of Immunization, Qinghai Provincial Center for Disease Control and Prevention, Xining 810007, China; akz1307@126.com; 2Department of Epidemiology, Key Laboratory of Public Health Safety, Ministry of Education, School of Public Health, Fudan University, Shanghai 200032, China; xylu20@fudan.edu.cn (X.L.); ljhu20@fudan.edu.cn (L.H.); bzli19@fudan.edu.cn (B.L.); 3Department of Health and Environmental Sciences, Xi’an Jiaotong-Liverpool University, Suzhou 215123, China; Jing.Wang18@student.xjtlu.edu.cn

**Keywords:** vaccine hesitancy, parental acceptance of childhood vaccination, childhood COVID-19 vaccine, resource-limited setting, minority population

## Abstract

China has initiated the COVID-19 vaccination for children aged 15–17 years since late July 2020. This study aimed to determine the association between adult vaccine hesitancy and parental acceptance of childhood COVID-19 vaccines in a multi-ethnicity area of northwestern China. A web-based investigation was performed with a convenience sampling strategy to recruit the parents aged 20–49 years. In a total of 13,451 valid respondents, 66.1% had received the COVID-19 vaccination, 26.6% were intent to receive, while 7.3% were not intent, with an increasing vaccine hesitancy (*p* < 0.001). Moreover, vaccination uptake of four common vaccines in their children remained low (29.0% for influenza vaccine, 17.9% for pneumonia vaccine, 10.9% for rotavirus vaccine, 8.0% for Enterovirus-71 vaccine), while overall parental acceptance of childhood COVID-19 vaccines was 50.0% (32.7% for those aged 0–5, 46.6% for 6–10, 73.3% for 11–18; *p* < 0.001). Vaccination uptake of these four vaccines and parental acceptance of childhood COVID-19 vaccine were negatively associated with adult vaccine hesitancy. In addition, respondents mostly preferred childhood COVID-19 vaccines with weak mild common adverse events (β = 1.993) and no severe adverse events (β = 1.731), demonstrating a positive association with adult vaccine hesitancy. Thus, it warrants specific countermeasures to reduce adult vaccine hesitancy and improve strategies for childhood vaccination.

## 1. Introduction

Vaccine hesitancy remains a public health challenge worldwide. It is defined as any delay in acceptance or refusal of vaccination, despite availability of vaccination services [1]. In recent years, vaccine hesitancy increased in both low- and middle-income countries and high-income countries, which made the entire population vulnerable to communicable diseases [2,3]. On one hand, adult vaccine hesitancy directly affects their vaccination behavior, such as low vaccination uptake (<49.0%) of flu vaccines in the United States and Europe [4,5]. In China, flu vaccine and 23-valent pneumococcal polysaccharide vaccine are free of charge in several cities; however, the acceptance remained low (<11.0%) [6,7]. On the other hand, adult vaccine hesitancy may further reduce parental intent to have children to be vaccinated. For instance, in Nigeria, adults’ boycott to polio immunization for children, driven by distrust, finally led to polio outbreaks across three continents [8]. Another example was the decline in childhood immunization of measles, mumps, and rubella (MMR) vaccine in Europe and the United States, which was caused by a concern of developing autism [9,10]. Previous studies in the United States and Canada found that experiences, emotions, routine ways of thinking, and risk perceptions and trust impact parental attitudes and decision-making processes of childhood immunization [2]. Thus, adult vaccine hesitancy may have a significant impact on the childhood immunization.

Currently, COVID-19 remains a pandemic all over the world. In Israel, two-dose COVID-19 vaccines have been evidenced to be highly effective across all age groups (≥16 years) in preventing SARS-CoV-2 infection and COVID-19-related hospitalization, severe disease, and death [11]. Notably, declines in incidence of COVID-19 corresponded with vaccination coverage, rather than nationwide lockdown [12]. It shed light on reopening the whole society. However, vaccine hesitancy has been documented since the implementation of COVID-19 vaccination [13]. The main concerns were efficacy and safety of COVID-19 vaccines, real need of COVID-19 vaccination, and social aspects such as cultural and religious beliefs, which varied across countries [14]. It has raised a new public health concern how to improve the confidence and intention of COVID-19 vaccination for achieving higher vaccination coverage.

Furthermore, the COVID-19 epidemic resurged in Israel, which might be attributable to the limited protection of COVID-19 vaccines against variants of SARS-CoV-2, and/or children becoming infected and spreading the virus in community [15]. It indicated a need of COVID-19 vaccination for children. We performed a study on the parental intention to vaccinate children against COVID-19 in an economically developed area of China in early 2020, which suggested parents’ concerns on the effectiveness of COVID-19 vaccines [16]. However, parental attitudes may have changed along with the pandemic and vaccination campaign. China has implemented a nationwide COVID-19 vaccination campaign in adults since March 2020 and in children aged 12–17 years since late July. Thus, we designed this study in Qinghai province in northwest China to determine the association between adult vaccine hesitancy and parental acceptance of COVID-19 vaccines for children, in order to improve strategies for childhood COVID-19 vaccination.

## 2. Materials and Methods

### 2.1. Study Design

Qinghai province is located in northwestern China, which is a multi-ethnicity living area and has relatively poor economic development. As of 31 July, 80.0% of the total population has received at least one dose of inactivated COVID-19 vaccines (BBIBP-CorV, Sinopharm, Beijing, China; and CoronaVac, Sinovac, Beijing, China), and 61.5% has received the second dose (full vaccination) in Qinghai province. We designed this study in children’s parents aged 20–49 years, with children 0–18 years. In China, children aged 18 years are at their final year of high school, so they have not been vaccinated against COVID-19 similar to Year 1 and 2 students and then included in this study. We excluded possible parents <20 years (as it is theoretically illegal to marry under 20 years old) and those >49 years (as they had a much lower fertility and might not be very informative).

As Qinghai province has a relatively large square (722,300 km^2^) and a limited population size (5.9 million), we utilized a web-based survey with a convenience sampling strategy, using Wenjuanxing (Questionnaire Star) that is a Chinese online platform facilitating questionnaire functions equivalent to Amazon Mechanical Turk [17]. The quick response (QR) code that accessed the web-based questionnaire was generated by the platform and then placed in a total of 385 community vaccination clinics and makeshift vaccination sites across Qinghai province in June and July 2020. In a pilot study of 20 children’s parents, we found that the minimum time for carefully completing the web-based questionnaire was 240 s, which was used as a threshold for screening the questionnaire records in this study.

### 2.2. Demographics

In the questionnaire, we investigated the respondents’ (parents) demographics, including their age and sex, number of children (single/two or more), ethnicity (Han/minority), educational level, registered residence (urban/rural), and monthly family income. In addition, we included the children’s age, which was defined as follows: (1) for single child, her/his age was included; and (2) for two or more children, the child whose age is between 0–17 years was included; if both children are 0–17 years old, the younger one was chosen.

### 2.3. Adult Vaccine Hesitancy

We utilized a 10-question adult vaccine hesitancy scale (aVHS) in this study. This scale had been validated by Chinese samples with a high internal reliability, as described elsewhere [18]. It measured general vaccine hesitancy and did not target on specific vaccines. Each question was assessed with a 5-point Likert scale. Negative attitude questions towards vaccine hesitancy were scored directly; for positive attitude questions, six points were subtracted from the questions’ scores so that all items had scores in the same direction. Subsequently, these scores were added together to obtain the total score of vaccine hesitancy.

### 2.4. COVID-19 Vaccination Behavior and Intent

We investigated the respondents’ vaccination status of COVID-19 vaccines and prepared four responses, including already vaccinated, not vaccinated but intend to vaccinate, not vaccinated in the near future, and not vaccinated and not intend to ever vaccinate. Subsequently, the last two responses were collapsed together, because none of them had a strong intention to vaccinate recently. We classified these responses into three groups: “already vaccinated with COVID-19 vaccines”, “intend to vaccinate”, and “not intend to vaccinate”.

Parental acceptance of COVID-19 vaccines for children included three responses as follows: intend to vaccinate, not intend to vaccinate, and unclear. The last two responses were also collapsed together. Two groups were then classified as “intend to have children vaccinated with COVID-19 vaccines” and “not intend to”.

### 2.5. History of Childhood Immunization and Pediatric Infectious Diseases

In China, there are two categories of vaccines for children. Category I vaccines, provided by the Government, are free of charge and mandatory. Category II vaccines are voluntarily chosen and paid by vaccinees at their own expense. This study selected four common Category II vaccines, including pneumonia, influenza, rotavirus and EV-71 vaccines, and investigated the vaccination experiences of these vaccines. We also included corresponding pediatric infectious diseases and history of adverse events following immunization (AEFI) in the questionnaire. Consequently, we explored the relationship between above items and parental acceptance of childhood COVID-19 vaccination.

### 2.6. Discrete Choice Experiment (DCE)

To explore the parental preferences for childhood COVID-19 vaccines, we designed a DCE. It was consisted of vaccine attributes and levels, including vaccine effectiveness (95% vs. 80%), mild common adverse events following immunization (AEFI) (1 day of headache and fatigue vs. 1–2 days of fever), rare but severe AEFI (no risk vs. same risk as flu vaccine), and number of doses (1 dose vs. 2 doses). All the attribute levels were designed according to the COVID-19 vaccines available in China, including inactivated vaccines and adenovirus-vectored vaccines [19]. Then we prepared a total of 16 choice sets with random combinations of attribute levels using a fractional factorial design, and then randomly allocated the sets into 4 blocks in the DCE. Each respondent randomly responded to 4 choice sets in each block and chose one of the two hypothetical childhood COVID-19 vaccines in each choice set [20].

### 2.7. Heat Map

The association between the intention to vaccinate their children against COVID-19 and vaccination status of four Category II vaccines was visualized in a heat map using R package (v3.6.3, R Foundation for Statistical Computing, Vienna, Austria). We calculated the Cramer’s V coefficient between two vaccines, which demonstrated the intensity of correlation: strong (the coefficient >0.6), moderate (0.3–0.6), or weak (<0.3).

### 2.8. Statistical Analysis

Demographics were compared across COVID-19 vaccination status among the respondents by using the Chi-square test. Vaccine hesitancy was calculated by demographic groups and COVID-19 vaccination groups and then compared using Kruskal–Wallis test. Furthermore, we explored the influencing factors associated with vaccination of four childhood vaccines and parental acceptance of childhood COVID-19 vaccines, by using logistic regression models. DCE data were analyzed using multinominal mixed logit regression models and predicted probability of preferences for childhood COVID-19 vaccines were calculated by hypothetical COVID-19 vaccines with combined vaccine attributes [20]. All the analysis was performed using R package (v3.6.3, R Foundation for Statistical Computing, Vienna, Austria). A *p* value of <0.05 was considered significant.

## 3. Results

### 3.1. Characteristics of Respondents

A total of 17,511 respondents completed the web-based questionnaire. We checked the respondents’ records and excluded those invalid due to completing the questionnaire less than 240 s (*n* = 1118), children’s parents <20 years old or >49 years old (*n* = 2631), or children >17 years old (*n* = 843). Finally, 13,451 respondents who had valid records were included for further analysis.

The respondents had an average age of 36.0 ± 5.7 years old. They were more likely to be woman (63.6%), rural residents (60.5%), with monthly family income less than CNY 7500 (68.1%), with education level of high school or below (71.5%), and have two or more children (64.6%). In addition, 31.5% of them were minority Chinese, which was much higher than the national average (8.5%) [21].

### 3.2. Adult Vaccine Hesitancy and COVID-19 Vaccination

Respondents expressed high agreement to the positive attitude questions in the aVHS, with that of “the information about vaccines is trustworthy” being lowest (81.3%) (Figure 1). Similarly, agreement to the negative attitude questions remained high, demonstrating their concerns of more risks of new vaccines (68.9%; those responded “neutral”, “agree”, and “strongly agree” for the L5) and little need of vaccines for uncommon diseases (70.5%; those responded “neutral”, “agree”, and “strongly agree” for the L10).

Moreover, 66.1% of respondents reported they had received the COVID-19 vaccination, 26.6% were intent to receive, while 7.3% were not intent to be vaccinated. Vaccine hesitancy increased with these three groups (*p* < 0.001) (Table 1). Respondents aged 40–49 years, female, Han Chinese, registered residence in urban areas, single child, higher educational level, and higher monthly family income were associated with a higher uptake of COVID-19 vaccine (each *p* < 0.001) (Table 1) and lower vaccine hesitancy (each *p* < 0.001).

### 3.3. Vaccination of Four Childhood Vaccines and Parental Acceptance of Childhood COVID-19 Vaccines

Vaccination uptake rate of four Category II vaccines in children was determined, including influenza vaccine (29.0%), pneumonia vaccine (17.9%), rotavirus vaccine (10.9%), and EV-71 vaccine (8.0%), among which there were moderate and strong correlations (Cramer’s V coefficients ranged between 0.4–0.7) (Figure 2). The overall parental acceptance of COVID-19 vaccines for children was 50.0% (6723/13,451) and had no correlation with above vaccination uptake. Furthermore, it was 32.7% (1486/4542) for children aged 0–5 years, 46.6% (2249/4830) for those aged 6–10 years, and 73.3% (2988/4079) for those aged 11–18 years, showing an increasing trend with children’ age groups (*p* < 0.001).

Vaccination uptake of above four vaccines and parental acceptance of childhood COVID-19 vaccines was negatively associated with adult vaccine hesitancy and differed across the demographic groups, especially children’s age, ethnicity, and educational level (Table 2). Remarkably, respondents whose children aged 0–5 years were most likely to have their children vaccinated with above four vaccines, whereas those who had older children were more intent to accept childhood COVID-19 vaccine. In addition, there were significant positive associations between medical history of influenza, pneumonia, and hand, foot and mouth disease, and uptake of corresponding vaccines; however, medical history of diarrhea was negatively associated with rotavirus vaccination. For parental acceptance of childhood COVID-19 vaccines, history of AEFI was a negative factor, while medical history of four infectious diseases did not have an association.

### 3.4. Parental Preferences for Childhood COVID-19 Vaccines

We estimated the respondents’ preferences for childhood COVID-19 vaccine attributes. Overall, respondents preferred a COVID-19 vaccine with 95% effectiveness (β = 1.138), one-day headache and fatigue as mild common AEFI (β = 1.993), no severe AEFI (β = 1.731), and one-dose schedule (β = 0.788) (each *p* < 0.001) (Table 3). Furthermore, we classified three subgroups by 25% and 75% percentiles of vaccine hesitancy score. Preference for weak mild common AEFI and no severe AEFI increased positively with vaccine hesitancy score (Table 3).

Regarding the relative importance of each vaccine attribute, respondents considered the form of mild common AEFI was the most important (35.6%; 95%CI: 32.3%, 38.0%), followed by rare but severe AEFI (30.9%; 95%CI: 27.4%, 33.6%), vaccine effectiveness (19.6%; 95%CI: 15.0%, 23.5%) and number of doses (14.0%; 95%CI: 10.0%, 17.6%). The most favored COVID-19 vaccine for children was which has 95% effectiveness, one-day headache and fatigue as mild common AEFI, no severe AEFI and one-dose schedule (93.7%; 95%CI: 93.0%, 94.5%). In contrast, the least favored vaccine was which has 80% effectiveness, 1–2 days of fever as mild common AEFI, same severe AEFI risk as flu vaccine and two-dose schedule (6.3%; 95%CI: 5.5%, 7.0%) (Figure 3).

## 4. Discussion

This study found that adult vaccine hesitancy was related to their own vaccination against COVID-19, and also associated with children’s uptake of existing vaccines and parental acceptance of childhood COVID-19 vaccines. It has been documented that children’s parents may have more influence on the decision-making of vaccination, compared to health care practitioners [22]. Similar findings were reported in the vaccination of measles and pertussis vaccines in the United States, regardless of active recommendation by health care practitioners [23]. A recent study in Wuxi city of eastern China has found that parental vaccine hesitancy led to a refusal to use COVID-19 and influenza vaccines [24]. In a previous study in Shanghai of eastern China in early 2020, we also found the parents were partially hesitant to have their children vaccinated with COVID-19 vaccines [16]. It has raised a significant public health concern how to tackle with adult vaccine hesitancy towards incoming childhood COVID-19 vaccination.

In this study, parents who are minority Chinese and have poor educational level reported lower uptake rate of existing childhood vaccines and also were hesitant to receive childhood COVID-19 vaccines. A recent review on context-specific causes of vaccine hesitancy in different settings examined the impact of parents’ educational level on vaccine hesitancy, which reported conflicting results. Studies in China, Lebanon, Israel, and the United States identified higher education as a potential barrier, whereas studies in Greece, the Netherlands, Nigeria, and Pakistan identified it as a promoter of vaccination [25]. In addition, our study was conducted in northwestern China that is a multi-ethnicity area with low urbanization and underdeveloped economics. In this study, the parental acceptance of childhood COVID-19 vaccines was determined to be 50%, and the uptake rate of four existing vaccines ranged between 8% and 29%, suggesting generally low acceptance of childhood vaccines. Previous studies in low- and middle-income countries revealed that regional, cultural, and economic factors played a significant role in vaccine hesitancy [26]. In addition, age may influence vaccine hesitancy [27,28]. We identified that older respondents were more likely to accept COVID-19 vaccines for themselves, whereas they did not differ in the intent to accept childhood vaccines with those younger respondents. Thus, further study on vaccine hesitancy and its influencing factors, especially in resource-limited settings and in minority population, is urgently warranted to address the disparity in the COVID-19 vaccination.

The COVID-19 vaccination has been the largest immunization campaign ever in history. It is currently accompanied by vaccine hesitancy, which may be attributable to a need of more vaccine information, anti-vaccine movement, or beliefs and a lack of trust [29]. Recently, a Phase 1/2 clinical trial of an inactivated COVID-19 vaccine developed by Sinovac Life Sciences (Beijing, China) has been proven to be safe and induce humoral responses in children aged 3–17 years [30]. In the DCE, the respondents highly expected weak mild common AEFI and no severe AEFI, demonstrating they remained concerned on safety for children, though the large COVID-19 immunization campaign had confirmed the safety of inactivated COVID-19 vaccines. In addition, preferences for these vaccine attributes differed across adult vaccine hesitancy, which implied that parental attitudes on childhood COVID-19 vaccines may correlate with vaccine hesitancy. Moreover, social, psychological, and behavioral factors have inevitable impacts on the promotion of a new childhood vaccine, in addition to vaccine attributes. For instance, parents thought vaccines are created and distributed for profit rather than disease prevention or believed diseases come and go in cycles that are not correlated with herd immunity [31]. In our study, parents whose children aged ≥6 years were more intent to accept childhood COVID-19 vaccines. It might indicate they had a better understanding of COVID-19 acquisition risk [32], or they just expressed the concerns of safety for younger children, as described elsewhere [33]. Thus, a “parental vaccine hesitancy scale for their children” may be designed for better assessment of vaccine hesitancy in these scenarios.

There are limitations in this study. As a web-based survey, we could not estimate the response rate. We placed the QR code in community vaccination clinics and makeshift vaccination sites, and health care practitioners (HCPs) helped encourage the children’s parents to scan the code and complete the questionnaire. However, HCPs could not always promote the parents to join the study, which may induce a bias. Furthermore, we recruited the children’s parents in 385 vaccination clinics and makeshift vaccination sites (totally 442 in Qinghai), which means it was very likely to recruit the respondents with higher intent to vaccinate themselves and their children with COVID-19 vaccines in our study. However, the respondents were from eight prefectural administrative regions (totally eight in Qinghai) and subsequently thirty-seven districts/counties (totally forty-four). Considering the sample size and sites of recruitment, our findings may be informative of adult vaccine hesitancy and parental acceptance of childhood vaccines. Another limitation was that we could not identify the temporal sequence between onset time of infectious diseases and vaccination time of corresponding vaccines. We did not investigate these points of time, as recall bias is very likely to occur for the respondents. Moreover, we could not directly determine a causal link between adult vaccine hesitancy and childhood vaccination. In this study, four common childhood vaccines were non-mandatory and self-paid, and COVID-19 vaccine was non-mandatory and free of charge, which may empower children’s parents for decision-making. However, it warrants more robust evidence.

There are also some novelty and strengths in this study. We determined possible disparity in the vaccination that might be overlooked in resource-limited settings and in minority population, in addition to previous studies focusing on the knowledge, awareness, and behavior of COVID-19 vaccination in China. Moreover, we quantitatively measured vaccine hesitancy and then we explored the association between the adult vaccine hesitancy and parental acceptance of childhood vaccination, which could rapidly improve the strategy for childhood COVID-19 vaccination.

## 5. Conclusions

Adult vaccine hesitancy remained moderate in northwestern China. It was negatively associated with vaccination uptake of four common Category II vaccines in children and parental acceptance of childhood COVID-19 vaccination. In addition, parents mainly expressed the concern on vaccine safety for a childhood COVID-19 vaccine, rather than effectiveness or number of doses, which was also positively associated with adult vaccine hesitancy. Our findings suggest that adult vaccine hesitancy may have a direct or indirect impact on childhood vaccination. It is of great significance to further reduce the parent’s vaccine hesitancy, which may facilitate improving strategies for childhood vaccination, especially in resource-limited settings and in minority population.

## Figures and Tables

**Figure 1 vaccines-09-01088-f001:**
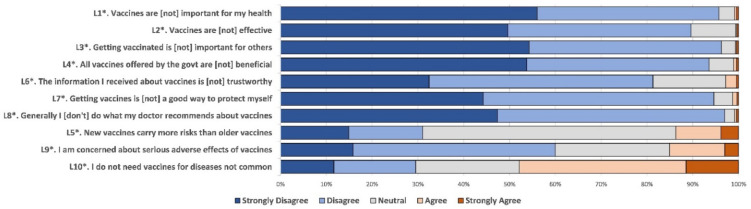
Responses to questions about vaccine hesitancy among the parents of children aged 0–17 years in Qinghai province of China, June–July 2021. Each question was assessed with a 5-point Likert scale. Questions with an asterisk (*) were positive attitude questions and then reversely coded so that all questions have responses with higher values being more vaccine hesitant.

**Figure 2 vaccines-09-01088-f002:**
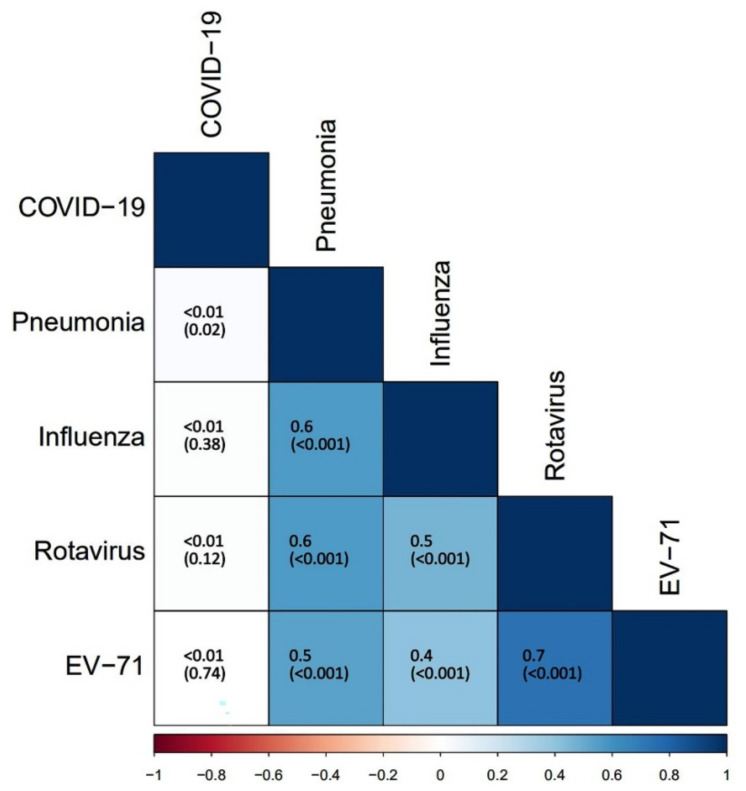
Correlation among vaccination uptake of four Category II vaccines in children and parental acceptance of childhood COVID-19 vaccines. The color in the lower left quarter represented the intensity of correlation. The Cramer’s V coefficients and *p* values in the parentheses were presented in the squares. Cramer’s V > 0.6 indicated a strong correlation, while Cramer’s V < 0.3 indicated a weak correlation.

**Figure 3 vaccines-09-01088-f003:**
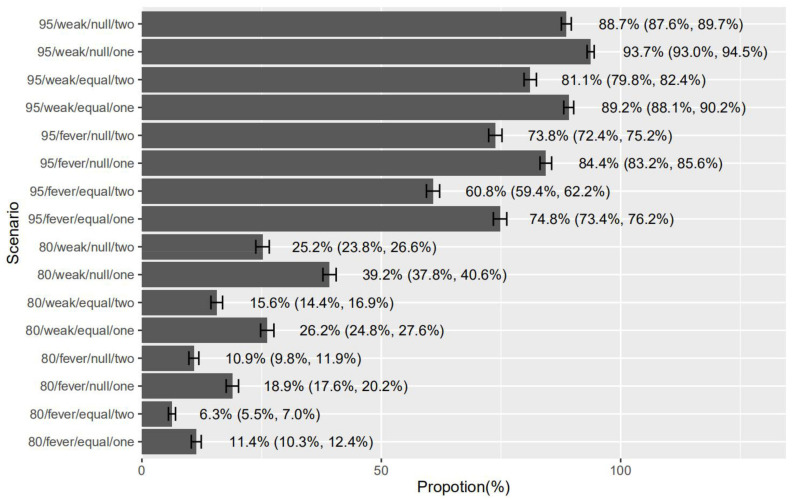
Predicted probability of preferences for childhood COVID-19 vaccines. 95: 95% vaccine effectiveness; 80: 80% vaccine effectiveness; weak: 1 day of headache and fatigue as mild common AEFI; fever: 1–2 days of fever as mild common AEFI; null: no severe AEFI; equal: the same risk of severe AEFI as flu vaccine; one: one-dose schedule; two: two-dose schedule.

**Table 1 vaccines-09-01088-t001:** Respondents’ demographics and adult vaccine hesitancy across their vaccination status of COVID-19 vaccines.

	Already Vaccinated with COVID-19 Vaccines (%)	Intend to Vaccinate COVID-19 Vaccines (%)	Not Intend to Vaccinate COVID-19 Vaccines (%)	*p* Value
Respondents’ age (years)				<0.001
20–29	1025 (60.8)	489 (29.0)	172 (10.2)	
30–39	5213 (66.6)	2018 (25.8)	593 (7.6)	
40–49	2657 (67.4)	1066 (27.0)	218 (5.5)	
Respondents’ sex				<0.001
Male	3202 (65.4)	1414 (28.9)	282 (5.8)	
Female	5693 (66.6)	2159 (25.2)	701 (8.2)	
Children’s age (years)				<0.001
0–5	3388 (74.6)	938 (20.7)	216 (4.8)	
6–10	3106 (64.3)	1363 (28.2)	361 (7.5)	
11–18	2400 (58.9)	1172 (28.7)	505 (12.4)	
Number of children				<0.001
Single	3306 (69.5)	1097 (23.1)	353 (7.4)	
Two or more	5589 (64.3)	2476 (28.5)	630 (7.2)	
Ethnicity				<0.001
Han Chinese	6456 (70.1)	2204 (23.9)	551 (6.0)	
Minority Chinese	2439 (57.5)	1369 (32.3)	432 (10.2)	
Educational level				<0.001
Middle school or below	3973 (58.0)	2286 (33.4)	593 (8.7)	
High school	1865 (67.6)	713 (25.8)	182 (6.6)	
College and university	3057 (79.6)	574 (15.0)	208 (5.4)	
Registered residence				<0.001
In urban areas	3975 (74.8)	1013 (19.1)	327 (6.2)	
In rural areas	4920 (60.5)	2560 (31.5)	656 (8.1)	
Monthly family income, CNY				<0.001
<5000	3392 (60.0)	1763 (31.2)	500 (8.8)	
5000–7499	2354 (67.2)	924 (26.4)	226 (6.4)	
≥7500	3149 (73.4)	886 (20.6)	257 (6.0)	
Adult vaccine hesitancy				<0.001
Mean ± SD	20.50 ± 4.33	21.21 ± 4.17	23.78 ± 4.39	

**Table 2 vaccines-09-01088-t002:** Vaccination acceptance of four childhood vaccines and intent to accept COVID-19 vaccine across respondents’ demographics by logistic regression.

		Odds Ratio (OR) Value (95% CI)				
	Number of Respondents	Pneumonia Vaccine	Influenza Vaccine	Rotavirus Vaccine	EV-71 Vaccine	Intent to Accept Child COVID-19 Vaccine
Children’s age (years)						
0–5	4542	1.0	1.0	1.0	1.0	1.0
6–10	4830	2.5 (2.1, 2.9) *	1.9 (1.7, 2.2) *	3.2 (2.7, 3.9) *	2.9 (2.4, 3.6) *	0.2 (0.1, 0.2) *
11–18	4079	1.8 (1.6, 2.1) *	1.6 (1.4, 1.8) *	1.9 (1.6, 2.2) *	1.6 (1.3, 2.0) *	0.3 (0.3, 0.3) *
Respondents’ age (years)						
20–29	1686	1.0	1.0	1.0	1.0	1.0
30–39	7824	1.0 (0.8, 2.0)	0.9 (0.8, 1.1)	1.2 (0.9, 1.5)	1.1 (0.8, 1.4)	1.0 (0.9, 1.2)
40–49	3941	1.0 (0.9, 1.2)	1.1 (0.9, 1.1)	1.2 (0.9, 1.4)	1.2 (0.9, 1.4)	0.9 (0.8, 1.0)
Number of children						
Single	4756	1.0	1.0	1.0	1.0	1.0
Two or more	8695	1.1 (0.9, 1.2)	1.0 (0.9, 1.1)	1.2 (1.1, 1.4) *	1.3 (1.2, 1.5) *	0.9 (0.9, 1.0)
Ethnicity						
Han Chinese	9211	1.0	1.0	1.0	1.0	1.0
Minority Chinese	4240	1.2 (1.1, 1.4) *	1.3 (1.2, 1.4) *	1.3 (1.2, 1.5) *	1.2 (1.0, 1.4) *	1.2 (1.1, 1.3) *
Educational level						
Middle school or below	6852	1.0	1.0	1.0	1.0	1.0
High school	2760	0.7 (0.6, 0.8) *	0.6 (0.5, 0.7) *	0.8 (0.7, 0.9) *	0.9 (0.8, 0.9) *	0.9 (0.8, 0.9) *
College and university	3839	0.9 (0.8, 0.9) *	0.8 (0.7, 0.9) *	0.9 (0.8, 0.9) *	0.9 (0.8, 0.9) *	0.9 (0.9, 0.9) *
Registered residence						
In urban areas	5315	1.0	1.0	1.0	1.0	1.0
In rural areas	8316	0.9 (0.7, 1.0)	0.9 (0.9, 1.0)	0.9 (0.8, 1.0)	0.9 (0.7, 1.0)	1.1 (1.0, 1.2) *
Monthly family income, CNY						
<5000	5655	1.0	1.0	1.0	1.0	1.0
5000–7499	3504	1.0 (0.9, 1.2)	0.9 (0.8, 1.0)	1.0 (0.9, 1.2)	1.2 (0.9, 1.4)	1.0 (0.9, 1.1)
≥7500	4292	1.1 (0.9, 1.2)	1.0 (0.9, 1.1)	0.9 (0.8, 1.1)	1.1 (0.9, 1.3)	1.0 (0.9, 1.2)
Medical history of pneumonia						
No	11,439	1.0	1.0	1.0	1.0	1.0
Yes	2012	0.8 (0.7, 0.9) *	0.8 (0.7, 0.9) *	0.9 (0.8, 1.1)	1.1 (0.9, 1.3)	1.0 (0.8, 1.1)
Medical history of influenza						
No	10,243	1.0	1.0	1.0	1.0	1.0
Yes	3208	0.8 (0.7, 0.9) *	0.6 (0.6, 0.7) *	0.9 (0.7, 1.0)	1.0 (0.8, 1.2)	0.9 (0.8, 1.0)
Medical history of diarrhea						
No	10,913	1.0	1.0	1.0	1.0	1.0
Yes	2538	1.1 (0.9, 1.2)	1.1 (0.9, 1.2)	1.4 (1.2, 1.7) *	1.1 (0.9, 1.4)	1.0 (0.9, 1.2)
Medical history of hand, foot and mouth disease						
No	12,470	1.0	1.0	1.0	1.0	1.0
Yes	981	0.9 (0.8, 1.1)	0.8 (0.7, 0.9) *	0.7 (0.6, 0.9) *	0.8 (0.6, 0.9) *	1.0 (0.8, 1.1)
History of AEFI **						
No/not sure	13,126	-	-	-	-	1.0
Yes	325	-	-	-	-	1.5 (1.2, 1.9) *
Adult vaccine hesitancy						
per incremental 1 score	-	0.9 (0.9, 0.9) *	0.9 (0.9, 0.9) *	0.9 (0.9, 0.9) *	0.9 (0.9, 0.9) *	0.9 (0.9, 0.9) *

* Variables that were statistically significant were marked in the table. ** History of AEFI was explored as a covariate with parental acceptance of child COVID-19 vaccines.

**Table 3 vaccines-09-01088-t003:** Mixed logit estimates and standard deviations with calculated proportions of positive effect for vaccine attributes.

	Vaccine Effectiveness 95% vs. 80%	Mild Common AEFI 1 Day of Headache, Fatigue vs. 1–2 Days of Fever	Rare but Severe AEFI No Risk vs. Same Risk as Flu Vaccine	No. Doses 1 Dose vs. 2 Doses
Total (*n* = 13,451)				
% Pos *	79.2	90.1	85.3	61.5
Estimate (SD)	1.138	1.993	1.731	0.788
SD	1.001	1.020	1.021	1.022
*p* value	<0.001	<0.001	<0.001	<0.001
Vaccine hesitancy score ≤18 (*n* = 4055)				
% Pos *	77.2	87.5	79.7	66.9
Estimate	1.137	1.889	1.675	0.827
SD	1.002	1.030	1.040	1.041
*p* value	<0.001	<0.001	<0.001	<0.001
Vaccine hesitancy score 19–24 (*n* = 6591)				
% Pos *	81.5	92.4	83.8	59.7
Estimate	1.140	2.070	1.724	0.759
SD	1.002	1.038	1.031	1.032
*p* value	<0.001	<0.001	<0.001	<0.001
Vaccine hesitancy score ≥25 (*n* = 2805)				
% Pos *	76.7	94.3	89.3	63.1
Estimate	1.135	2.136	1.835	0.798
SD	1.003	1.046	1.048	1.049
*p* value	<0.001	<0.001	<0.001	<0.001

* % Pos was the proportion of respondents who had a positive preference for a vaccine attribute.

## Data Availability

The datasets generated during the current study are not publicly available due to privacy but are available from the corresponding author Yihan Lu on reasonable request.

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
