# Peer review of "Association between Adult Vaccine Hesitancy and Parental Acceptance of Childhood COVID-19 Vaccines: A Web-Based Survey in a Northwestern Region in China"

_vaccines, 2021, doi:10.3390/vaccines9101088_

Round 1

Reviewer 1 Report

This survey aimed at assessing the relationship between vaccine hesitancy and the parental acceptance of COVID-19 vaccines for their children in a region of north-eastern China. The subject is of great interest and the manuscript is generally clear and well written. However, some minor points should be addressed.

Lines 114-115. Could you clarify this point? What do you mean?

Line 189. Please provide the full name of aVHS.

Lines 191-192. The percentages reported in the text seem different from those illustrated in Figure 1 (I suppose questions L9 and L10).

Lines 214-216. In which table/figure have these data been shown?

Do the odd ratios in Table 2 refer to acceptance or hesitation towards vaccines? In other words, does OR> 1 mean vaccination hesitation? Please clarify this point by rewording the table title accordingly.

Line 249. Consider adding “the” before “most”.

Line 287. Please correct the typo in “revealed”.

Line 298. Consider deleting “that” before “had”.

Author Response

Reviewer 1

  1. Lines 114-115. Could you clarify this point? What do you mean?

Authors’ response: Thank you for your suggestion. We include both positive and negative attitude questions towards vaccine hesitancy in the 10-question adult Vaccine Hesitancy Scale (aVHS); each question is assessed with a 5-point (1, 2, 3, 4, and 5) Likert scale. In order to calculate a sum score of vaccine hesitancy, we define that negative attitude questions were scored directly, and positive questions were scored by subtracting by 6 (6-1=5, 6-2=4, 6-3=3, 6-4=2, and 6-5=1), to ensure the same direction of positive question scores with that of negative question scores.

  1. Line 189. Please provide the full name of aVHS.

Authors’ response: Thank you for your comments. We have mentioned the adult Vaccine Hesitancy Scale (aVHS) in the Materials and Methods for the first time and then used the abbreviation aVHS in the following text.

  1. Lines 191-192. The percentages reported in the text seem different from those illustrated in Figure 1 (I suppose questions L9 and L10).

Authors’ response: Thank you for your detailed comments. “Concerns of more risks of new vaccines (68.9%)” was the response for the question L5, and “Little need of vaccines for uncommon diseases (70.5%)” was for the question L10. Both percentages were the proportion of the respondents who responded “neutral”, “agree”, and “strongly agree”. We have revised accordingly to clarify it as follows:

“Similarly, agreement to the negative attitude questions remained high, demonstrating their concerns of more risks of new vaccines (68.9%; those responded “neutral”, “agree”, and “strongly agree” for the L5) and little need of vaccines for uncommon diseases (70.5%; those responded “neutral”, “agree”, and “strongly agree” for the L10).” (Lines 195-199).

  1. Lines 214-216. In which table/figure have these data been shown?

Do the odd ratios in Table 2 refer to acceptance or hesitation towards vaccines? In other words, does OR> 1 mean vaccination hesitation? Please clarify this point by rewording the table title accordingly.

Authors’ response: Thank you for your detailed comments. These data were not shown in any table or figure in the manuscript. We have added the denominators and numerators for these data (Lines 219-223).

In addition, the OR> 1 refers to the acceptance towards vaccines (four childhood vaccines) and intent to accept (COVID-19 vaccine) in Table 2. We have revised the title of Table 2 to “Vaccination acceptance of four childhood vaccines and intent to accept COVID-19 vaccine across respondents’ demographics by logistic regression”.

  1. Line 249. Consider adding “the” before “most”.

Authors’ response: Thank you for your comments. We have added it in the manuscript (Lines 256).

  1. Line 287. Please correct the typo in “revealed”.

Authors’ response: Thank you for your comments. We have corrected it in the manuscript (Lines 294).

  1. Line 298. Consider deleting “that” before “had”.

Authors’ response: Thank you for your comments. We have deleted it in the manuscript (Lines 309).

Reviewer 2 Report

I was invited to revise the paper entitled "Association between adult vaccine hesitancy and parental acceptance of childhood COVID-19 vaccines: A web-based survey in a northwestern region in China". The topic is interesting and the study was well conducted. It can improve the knowledge of the field about this topic. I have some minor observations:

  • Scores are discrete varibles so Authors can't use parametric tests, such ANOVA, to compare scales among study groups;
  • In table 2 Authors should also report the number of patients for each variables. Please add a column for each variable;
  • Discussions should highlight the impact of age on vaccine hesitancy, as reported by 10.1016/j.socscimed.2014.04.018 and 10.3390/vaccines8020248  

Author Response

Reviewer 2

  1. Scores are discrete variables so Authors can't use parametric tests, such ANOVA, to compare scales among study groups.

Authors’ response: Thank you for your valuable comments. We have revised the statistical method to “Kruskal-Wallis test” in the Materials and Methods, and revised accordingly in the Results in the manuscript.

  1. In table 2 Authors should also report the number of patients for each variables. Please add a column for each variable.

Authors’ response: Thank you for your detailed comments. We have added a column “Number of respondents” for each variable in Table 2.

  1. Discussions should highlight the impact of age on vaccine hesitancy, as reported by 10.1016/j.socscimed.2014.04.018 and 10.3390/vaccines8020248  

Authors’ response: We appreciate your valuable suggestion. We have added more discussion and cited these two papers in the Discussion.

Reviewer 3 Report

General comments

This is a timely study on the ongoing prospects concerning the COVID-19 vaccination in adults and age groups of children concerning the uptake of other childhood vaccines, and parental acceptance and hesitancy concerning the minors’ vaccination rate. This web-based study encompassed more than 13000 valid responses and was performed in China. The general conclusions are that the higher the COVID-19 vaccination uptake (implicitly in parents) and parental acceptance, the lower is the parental vaccine hesitancy for COVID-19 (implicitly concerning their children, I guess).

The above conclusions visible in the abstract should be written more clearly. The problem also is that the parents’ self-acceptance of COVID-19 vaccination, could be hardly linked to greater hesitancy for the same vaccination in their children. Although theoretically possible, it would be highly unlikely in practice. Therefore, this conclusion is a kind of trite confirming the obvious.

Further, the authors point out that multi-ethnic Chinese minorities, often of poor socioeconomic status are prone to have greater hesitancy for COVID-19 vaccination, which translates into a lower rate concerning other childhood vaccinations. Interestingly, parental acceptance of childhood COVID-19 vaccinations increases with increasing age groups being the highest in late childhood/early adolescence. Although the study is timely and might be useful, I have some more polemic remarks as per specific comments.

Specific comments.

1/ Abstract is written in a complex and difficult to comprehend way. It should be simplified. You do not have to mention all aspects and all information in the abstract. Besides, I have a problem with separating the groups of adult (parental or not for that matter?) ethnic Chinese minorities from who - ‘true’ Chinese (sounds discriminatory). In the second part of the abstract, the demarcation line between these two groups gets somehow lost.

2/ To me, a more serious problem appears with your conclusions. That quote/unquote “adult vaccine hesitancy affects routine childhood immunization ….”, which is the main conclusion, is a kind of trite. Would you expect something opposite to be? By the way, is it adult vaccine hesitancy concerning own adult vaccination or directed specifically to children? It seems that these two things may be different, i.e., an adult may be hesitant about self-vaccination but not that of his children, or vice versa, or both. You can complicate the matter further, but better get it disentangled.

3/ Introduction offers a nice and useful survey of SARS-Cov-2 infections, hospitalizations, and vaccinations across continents. However, in your study aim …’ adult vaccine hesitancy and parental acceptance of COVID-19 vaccines for children….’ You mean adult vaccine hesitancy as the general aptitude or philosophic point of view or specifically directed to own children. The two are not necessarily the same.

4/ Did you happen to survey adults who do not have children concerning their vaccine hesitancy, or you discarded such respondents?

5/ The methods, surveys, and statistical elaboration do not raise any reservations. However, just of curiosity, I wonder why the population sample is rather small for a web-based study in a huge province with millions of inhabitants – I would have expected a much bigger cohort.

6/ Results are well presented in graphical and tabulated forms. In Table 1, parentheses refer to percentages, I guess. That should better be marked and is not.

7/ My opinion is that thousandths of values, for instance in odds ratio and confidence intervals in Table 2 and other occasions are unneeded in this kind of survey research. It would read much better with decimals; same concerning the hundredths of percentage values in Fig. 3.

8/ The discussion section is well designed. I like the idea of creating a scale for adult vaccine hesitancy. Whatever it would look like, it would be something new on the matter. However, you do not go beyond an empty proposal and drop this thread immediately.

I appreciate the extent and certainly burden of this work. However, the biggest problem I have with this text is that it contains many mutually entangled threads. The conclusions are unclear or kind of self-perpetuating in the sense they cannot expectedly be much different. We know all that, don’t we? It would be much helpful if you could finish the paper by describing its novelty and strengths.

Author Response

Reviewer 3

General comments

  1. This is a timely study on the ongoing prospects concerning the COVID-19 vaccination in adults and age groups of children concerning the uptake of other childhood vaccines, and parental acceptance and hesitancy concerning the minors’ vaccination rate. This web-based study encompassed more than 13000 valid responses and was performed in China. The general conclusions are that the higher the COVID-19 vaccination uptake (implicitly in parents) and parental acceptance, the lower is the parental vaccine hesitancy for COVID-19 (implicitly concerning their children, I guess). The above conclusions visible in the abstract should be written more clearly.

Authors’ response: Thank you for your detailed comments. We have revised accordingly in the Abstract.

  1. The problem also is that the parents’ self-acceptance of COVID-19 vaccination, could be hardly linked to greater hesitancy for the same vaccination in their children. Although theoretically possible, it would be highly unlikely in practice. Therefore, this conclusion is a kind of trite confirming the obvious.

Authors’ response: Thank you for your valuable comments. In China, a nationwide COVID-19 immunization program has been implemented in adults since March 2021, and then initiated in children aged 12-17 years since July. We have observed obvious vaccine hesitancy in both adults and their children. Thus, we had a hypothesis that adult vaccine hesitancy may further result in childhood vaccine hesitancy. Considering the coverage of full COVID-19 vaccination remained 61.5% in adults on July 31, we decided to determine possible association between the adult vaccine hesitancy and parental acceptance of childhood COVID-19 vaccine. In our study, four common childhood vaccines were non-mandatory and self-paid vaccines, and COVID-19 vaccine was non-mandatory and free of charge, which may empower children’s parents for decision-making. Thus, it would be possible that adult vaccine hesitancy may have an impact on the childhood vaccination. In addition, we have added more discussion on the limitations (Lines 337-348).

  1. Further, the authors point out that multi-ethnic Chinese minorities, often of poor socioeconomic status are prone to have greater hesitancy for COVID-19 vaccination, which translates into a lower rate concerning other childhood vaccinations. Interestingly, parental acceptance of childhood COVID-19 vaccinations increases with increasing age groups being the highest in late childhood/early adolescence. Although the study is timely and might be useful, I have some more polemic remarks as per specific comments.

Authors’ response: Thank you for your detailed comments. Respondents being minority Chinese and with lower socioeconomic status were more likely to have higher vaccine hesitancy, suggesting it may be crucial to further address the disparity in resource-limited settings and in minority population. In addition, we found that parental acceptance of childhood COVID-19 vaccinations increased with increasing age groups, which may support a step-by-step immunization strategy starting from early adolescence to late childhood and claim the transparency of robust evidence confirming the effectiveness and safety of the COVID-19 vaccines.

Specific comments.

  1. Abstract is written in a complex and difficult to comprehend way. It should be simplified. You do not have to mention all aspects and all information in the abstract. Besides, I have a problem with separating the groups of adult (parental or not for that matter?) ethnic Chinese minorities from who - ‘true’ Chinese (sounds discriminatory). In the second part of the abstract, the demarcation line between these two groups gets somehow lost.

Authors’ response: We appreciate your valuable comments. We have revised accordingly for more clarification in the Abstract. As we performed the study in a multi-ethnicity area, we included the ethnicity as a variable in the analysis. It was determined to be associated with adult vaccine hesitancy, whereas was not associated with vaccination acceptance of four common childhood vaccines or parental intent to accept childhood COVID-19 vaccine, so it was not presented in the second part of the Abstract.

In addition, we are so sorry for your discomfort. In this study, we classified ethnic minority Chinese and Han Chinese, which are internationally certified names. We declare there is no discrimination towards certain sociodemographic groups.

  1. To me, a more serious problem appears with your conclusions. That quote/unquote “adult vaccine hesitancy affects routine childhood immunization ….”, which is the main conclusion, is a kind of trite. Would you expect something opposite to be? By the way, is it adult vaccine hesitancy concerning own adult vaccination or directed specifically to children? It seems that these two things may be different, i.e., an adult may be hesitant about self-vaccination but not that of his children, or vice versa, or both. You can complicate the matter further, but better get it disentangled.

Authors’ response: Thank you for your valuable suggestion. In this study, we decided to determine possible association between the adult vaccine hesitancy and parental acceptance of childhood COVID-19 vaccine. As the four common childhood vaccines were non-mandatory and self-paid vaccines, and COVID-19 vaccine was non-mandatory and free of charge, children’s parents could make decision on their children’s vaccination. Thus, it would be possible that adult vaccine hesitancy may have an impact on the childhood vaccination. We have revised accordingly for more clarification in the Conclusions (Lines 353-359).

  1. Introduction offers a nice and useful survey of SARS-Cov-2 infections, hospitalizations, and vaccinations across continents. However, in your study aim …’ adult vaccine hesitancy and parental acceptance of COVID-19 vaccines for children….’ You mean adult vaccine hesitancy as the general aptitude or philosophic point of view or specifically directed to own children. The two are not necessarily the same.

Authors’ response: Thank you for your detailed comments. In this study, we measured adult vaccine hesitancy that is general and not targeted on specific vaccines, using a scale tool described elsewhere (Modification of a vaccine hesitancy scale for use in adult vaccinations in the United States and China. Hum Vaccin Immunother, 2021). Then we designed the questions for the parents what decision they would make on their own children’s COVID-19 vaccination. We have revised accordingly in the Materials and Methods (Lines 116-117).

  1. Did you happen to survey adults who do not have children concerning their vaccine hesitancy, or you discarded such respondents?

Authors’ response: Thank you for your comments. We designed this study to recruit the participants aged 20-49 years, with children 0-18 years. Thus, there is no respondent who do not have children in this study.

  1. The methods, surveys, and statistical elaboration do not raise any reservations. However, just of curiosity, I wonder why the population sample is rather small for a web-based study in a huge province with millions of inhabitants – I would have expected a much bigger cohort.

Authors’ response: Thank you for your comments. As Qinghai province has a relatively large square (722,300 km2) and a limited population size (5.9 million), we utilized a web-based survey with a convenience sampling strategy. As you said, this is a timely study, so we performed the study between June and July. In addition, we placed the QR code in community vaccination clinics and makeshift vaccination sites, and health care practitioners (HCPs) helped encourage the children’s parents to scan the code and complete the questionnaire. However, HCPs could not always promote the parents to join the study. Thus, we obtained a large sample size compared to similar web-based studies, while a small size compared to the whole population in the province. We have revised accordingly in the limitations (Lines 323-326, 342-348).

  1. Results are well presented in graphical and tabulated forms. In Table 1, parentheses refer to percentages, I guess. That should better be marked and is not.

Authors’ response: Thank you for your detailed comments. We have added it in Table 1.

  1. My opinion is that thousandths of values, for instance in odds ratio and confidence intervals in Table 2 and other occasions are unneeded in this kind of survey research. It would read much better with decimals; same concerning the hundredths of percentage values in Fig. 3.

Authors’ response: Thank you for your detailed comments. We have revised accordingly in Table 2 and Figure 3.

  1. The discussion section is well designed. I like the idea of creating a scale for adult vaccine hesitancy. Whatever it would look like, it would be something new on the matter. However, you do not go beyond an empty proposal and drop this thread immediately. I appreciate the extent and certainly burden of this work. However, the biggest problem I have with this text is that it contains many mutually entangled threads. The conclusions are unclear or kind of self-perpetuating in the sense they cannot expectedly be much different. We know all that, don’t we? It would be much helpful if you could finish the paper by describing its novelty and strengths.

Authors’ response: Thank you for your valuable suggestion. There are multiple limitations to this study. In contrast, we also have some novelty and strengths. This study determined possible disparity in the vaccination that might be overlooked in resource-limited settings and in minority population, in addition to previous studies focusing on the knowledge, awareness, and behavior of COVID-19 vaccination in China. Furthermore, we quantitatively measured vaccine hesitancy. More important, we explored the association between the adult vaccine hesitancy and parental acceptance of childhood vaccination, which could rapidly improve the strategy for childhood COVID-19 vaccination. We have revised accordingly in the Discussion and Conclusions (Lines 337-348, 350-359).

Round 2

Reviewer 2 Report

Authors modified the paper accordingly to suggested reviews. The paper is now acceptable for publication.

Author Response

Authors modified the paper accordingly to suggested reviews. The paper is now acceptable for publication.

Authors’ response: We really appreciate the reviewer’s comments.

Reviewer 3 Report

The body text of this manuscript is now improved. However, you could do a little more.

In the abstract, all acronyms, e.g., EV-71 and AEFI should be explained in full words at the first appearance.

This sentence in the discussion “In addition, age may play an impact on the vaccine hesitancy 291 [27,28]” is confusing in English. ‘impact’ is slangy and have a definite negative connotation. Thus, you suggest that age decreases the intent, and your result is the opposite. Why not use the nice English word ‘influence’ instead; ‘age may influence vaccine hesitancy’.

Author Response

The body text of this manuscript is now improved. However, you could do a little more. In the abstract, all acronyms, e.g., EV-71 and AEFI should be explained in full words at the first appearance.

Authors’ response: Thank you for your detailed comments. We have revised accordingly in the Abstract.

This sentence in the discussion “In addition, age may play an impact on the vaccine hesitancy 291 [27,28]” is confusing in English. ‘impact’ is slangy and have a definite negative connotation. Thus, you suggest that age decreases the intent, and your result is the opposite. Why not use the nice English word ‘influence’ instead; ‘age may influence vaccine hesitancy’.

Authors’ response: Thank you for your detailed comments. We have revised accordingly in the manuscript.